Biomechanical determinants of high ball speed during instep soccer kick by prepubescent male athletes: the importance of muscle elasticity

Deniz Volkan 1 volkand2014@gmail.com
Kılcı Abdullah 2
1 Department of Physiotherapy and Rehabilitation, Tarsus University , Mersin , Turkey
2 Department of Sport Sciences, Çukurova University , Adana , Turkey
Vieira Marcus
Electronic publication date: 2025 Aug 13
Publication date: 2025
Volume: 13
Electronic Location ID: e19923
Received 2025 Apr 11; Accepted 2025 Jul 24
Copyright: © 2025 Deniz and Kılcı
Copyright year: 2025
Copyright holder: Deniz and Kılcı
License: This is an open access article distributed under the terms of the Creative Commons Attribution License, which permits unrestricted use, distribution, reproduction and adaptation in any medium and for any purpose provided that it is properly attributed. For attribution, the original author(s), title, publication source (PeerJ) and either DOI or URL of the article must be cited.
License URL: https://creativecommons.org/licenses/by/4.0/

Keywords: Instep kick, Ball speed, Muscle elasticity, Muscle activation, Leg angular velocity

Funding: The authors received no funding for this work.

==============================
Background

The purpose of this study was to determine whether muscle viscoelastic properties, muscle activation and thigh rotational velocity and rotational acceleration are significantly associated with high ball speed during instep soccer kick by prepubescent male athletes.

Methodology

This study included 31 prepubescent male soccer players. Maximal ball speed during the instep kick was measured using a radar gun. Viscoelastic properties such as tone, elasticity, and stiffness of the superficial abdominal and leg muscles were measured at rest using myotonometry. The activation of the rectus abdominis (RA) and rectus femoris (RF), as well as the thigh’s three-dimensional rotational velocity and acceleration, were evaluated using surface electromyography and an inertial measurement unit.

Results

Significant negative correlations were found between the maximum ball speed and the hamstring medialis (HM) and gastrocnemius medialis (GM) elasticity (r = −0.656 and −0.680; power > 0.95; p < 0.05 for all). Significant positive correlations existed between maximum ball speed and RA–RF activation (r = 0.494 and 0.579), maximum thigh rotational velocity in the sagittal plane (r = 0.619), and maximum thigh rotational acceleration in the sagittal (r = 0.435) and horizontal (r = 0.380) planes (power > 0.95; p < 0.05 for all). The multiple linear regression analysis demonstrated that significant parameters for maximum ball speed were HM (B = −36.84) and GM (B = −26.83) elasticity along with thigh rotational velocity in sagittal plane (B = 0.01) (adjusted R2 = 0.56, delta R2 = 0.17).

Conclusions

The elasticity of the GM and HM muscles, the activation levels of the RA and RF muscles, and the rotational velocity-acceleration of the thigh in the sagittal and horizontal planes were significant factors associated with high ball speed during the instep kick. To enhance ball speed during the instep kick, training methods that focus on improving the elasticity of the GM and HM muscles, as well as increasing activation of the core and rectus femoris, should be prioritized for prepubescent male soccer players.

Introduction

The instep kick is the most powerful and widely used technique in soccer, particularly when high ball velocity is required. It is executed by striking the ball with the dorsum of the foot while the ankle is plantarflexed, allowing for maximal force transfer to achieve long-distance and high-speed trajectories (Lees & Nolan, 1998; Amiri-Khorasani & Ferdinands, 2014). The effectiveness of soccer kicking is determined by both the velocity and accuracy of the ball (Lees & Nolan, 1998). Although accuracy is significant, assessments of soccer kicking performance have primarily focused on maximizing ball speed (Lees & Nolan, 1998; Markovic, Dizdar & Jaric, 2006). Provided that the kick is precise, a higher ball speed enhances the likelihood of scoring by reducing the time for the goalkeeper to react (Markovic, Dizdar & Jaric, 2006). Numerous studies have examined the function of ball speed in soccer by determining the elements that influence the maximum kicking velocity. These factors encompass various aspects such as limb dominance (Campo et al., 2009), technique (Lees & Nolan, 1998), approach speed and angle (Kellis, Katis & Gissis, 2004), skill level (Cometti et al., 2001), lower extremity kinematics (Juárez et al., 2011), muscle strength and power (Manolopoulos, Papadopoulos & Kellis, 2006), and muscle activity (Cerrah et al., 2011; Shinkai et al., 2009). These researches have focused on highlighting the influence of the trunk and lower limb muscles on kicking performance. Given the growing body of recent studies on kicking biomechanics, the need for incorporating muscle viscoelastic properties into the analysis should be reassessed, particularly in studies on prepubescent soccer athletes.

The viscoelastic properties of muscles, including tone, elasticity, and stiffness, influence the alignment of body segments and their ability to adapt to movement (Creze et al., 2019). The viscoelastic properties of muscles exhibit significant variability influenced by age, sex, and sport-specific adaptations, reflecting the complex interplay between biological and functional factors in musculoskeletal dynamics. A recent biomechanical study revealed that the viscoelastic properties of scapulothoracic muscles significantly impact joint kinematics, including range of motion and rotational velocity, during high-speed upper extremity movements, irrespective of athletes’ physical and sport-specific characteristics (Deniz et al., 2024). From a biomechanical perspective, it can be hypothesized that the viscoelastic properties of lower extremity muscles may have a critical influence on the stretching–shortening cycle of the leg muscles. The stretching–shortening cycle of the leg muscles in the backswing and acceleration phases of the instep kick may be a considerable biomechanical factor affecting ball speed (Amiri-Khorasani, Osman & Yusof, 2011). In this cycle, the ability of the anterior muscles to stretch with controlled eccentric contraction in the stretching phase affects the concentric contraction of the anterior leg muscles in the shortening phase. Likewise, the ability of the gluteus maximus, hamstrings, and gastrocnemius to stretch with optimal eccentric contraction during the acceleration phase of the kick may be a determinant of the ball speed (Amiri-Khorasani, Osman & Yusof, 2011). While previous studies have focused on muscular strength (Manolopoulos, Papadopoulos & Kellis, 2006), muscle activation (Cerrah et al., 2011), and joint kinematics during kicking (Brophy et al., 2007; Cerrah et al., 2024) the role of muscle elasticity—particularly in prepubescent athletes—has remained underexplored despite its clear biomechanical relevance.

This study primarily aimed to examine the relationship between maximum ball speed and the viscoelastic properties of the superficial abdominal and leg muscles during the instep kick performed by prepubescent soccer players. It also aimed to investigate the association between abdominal and thigh muscle activity, as well as the maximum rotational velocities and accelerations of the kicking thigh in the sagittal, horizontal, and frontal planes with maximal ball speed.

The hypotheses of the present study were as follows: (i) There is a positive correlation between the viscoelastic properties of lower extremity muscles and maximum ball speed during instep soccer kick. (ii) There is a positive association between rectus abdominis (RA) and rectus femoris (RF) muscle activation, as well as thigh rotational velocity and acceleration, and maximum ball speed during instep kick performed by prepubescent male athletes.

Methods

Participants and instep kick

Portions of this text were previously published as part of a preprint (Deniz & Kilci, 2024). This observational, cross-sectional study was conducted at Atakent Sports Club (Adana, Türkiye). Prepubescent male soccer players with official licenses issued by the Turkish Football Federation were recruited. All 11- to 12-year-old athletes registered with Atakent Sports Club and their legal guardians were invited to participate in the study through text messages and verbal communication. All athletes who agreed to participate were included. A total of 37 players were initially enrolled, and data from 31 participants who met the inclusion criteria were collected and analyzed. All participants were in the prepubertal stage, as determined through observational screening by researchers. Players from all positions were included in the tests, except goalkeepers due to the distinct nature of their role. The players had an average playing experience of 28.7 ± 11.2 months, ranging from 18 to 60 months. The baseline characteristics of the athletes are shown in Table 1. All participants were injury-free and were preparing for competition at the time of data collection. This study was approved by the Clinical Research Ethics Committee of Cukurova University (Decision no.: 143/75, Approval date: 5 April 2024). All athletes were informed about the study protocol, and written consent was obtained from the participants and their legal representatives.

Table 1 Baseline characteristics of the players.

Variable	Mean ± SD	Minimum–maximum	
Age (years)	11.7 ± 0.5	11.0–12.0	
Height (cm)	143.3 ± 9.0	130.0–160.0	
Weight (kg)	37.5 ± 6.3	27.0–58.0	
Body mass index (kg/m2)	18.2 ± 1.6	15.9–22.9	
Number of months played soccer	28.7 ± 11.3	18.0–60.0	
Note:

SD, Standard deviation.

All testing was completed in a soccer training field with natural grass surface to represent competition performance and provide athletes with maximum freedom of movement. On the days the data were collected (23 and 25 April 2024), ambient temperature and humidity levels were monitored using a thermometer and a hygrometer located at the sports club where the testing was conducted. On 23 April, the temperature during testing ranged between 35–36 °C, with a relative humidity of approximately 36%. On 25 April, the temperature ranged between 33–34 °C, and the relative humidity was approximately 38%. Wind speed data were obtained from the Turkish State Meteorological Service’s Adana Regional Directorate station, located approximately 5 km from the test site. According to these records, the maximum wind speed was 4 km/h on both days. Additionally, observationally, the weather was calm with minimal wind, ensuring stable atmospheric conditions for testing. The task performed was an instep kick at maximum speed on a stationary ball. The standard-sized (Lotto FB1000 4, Trevignano, Italy) and standard-inflated (11 psi) ball was kicked 11 m toward a 2 × 2 m goal (Cerrah et al., 2024). During the preliminary trials, some variation among participants in their approach angles and run-up distances was observed. To control for these potential confounders and reduce inter-individual variability, these parameters were standardized based on the most commonly preferred values observed across participants. Specifically, players were instructed to use a run-up angle between 30° and 45° and to take approximately 2–3 steps—an approach consistent with those reported in recent studies involving youth soccer players (Augustus, Hudson & Smith, 2024; Cerrah et al., 2024).

Before the task, the players performed a standardized warm-up consisting of 10-min of submaximal running and several whole-body stretching exercises. They then performed five submaximal instep soccer kicks. Following this warm-up, the athletes were asked to perform an instep kick with maximum ball speed, and they were required to try to hit the target area. Each player performed three instep kicks with their dominant leg, and the muscle activation and thigh rotational velocity during the successful kick that produced the highest ball speed were analyzed. (Dörge et al., 1999; Kellis, Katis & Gissis, 2004).

Outcome measures

Maximum ball speed

Ball exit velocity (i.e., maximal ball speed) was measured using a radar gun (Bushnell Velocity Speed Gun, Overland Park, KS, USA), which is capable of measuring speeds ranging from 16 to 177 km/h at distances up to 27 m, with an accuracy of ±2 km/h. This radar gun has high reliability (intraclass correlation coefficient = 0.91) for measuring ball speed (Přidal, Matušov & Mikulič, 2023). The radar device was positioned by an observer 1.5 m behind the goal and at a height of 1.5 m, directly aligned with the ball’s trajectory to ensure a clear line of sight (Ferraz, van den Tillaar & Marques, 2017) (Fig.1).

Figure 1 Set up of instep kicking motion and devices.

Myotonometry

Myotonometry provides objective data on the viscoelasticity of muscles, such as tone, elasticity, and stiffness. Tone characterizes the intrinsic tension of biological soft tissues on the cellular level. Stiffness characterizes the resistance of biological soft tissues to a force of elongation or deformation. Elasticity is the biomechanical property of soft tissues that characterizes the ability of elongation and to recover its initial shape from being deformed (Pruyn, Watsford & Murphy, 2016). It uses a smartphone-sized, handheld device for myotonometric measurement (MyotonPro®, Myoton AS, Tallinn, Estonia). Myotonometry has shown good to excellent reliability in evaluating the viscoelastic properties of muscles in childeren (Lidström et al., 2009). The MyotonPro® device determines the tone, elasticity, and stiffness values of the muscles by analyzing the characteristics of the oscillation wave formed in response to five short (0.15 ms) stimuli applied by the probe to the tissue with a force of 0.56 N and a frequency of 1 Hz (Chang et al., 2024; Deniz et al., 2024).

The viscoelastic properties of muscles engaged in the stretch-shortening cycle during the instep kick were assessed. Myotonometric assessment was performed of the major superficial and having minimal overlying adipose tissue muscles in the abdomen and lower extremity: the RA, RF, gluteus medius, adductor magnus, hamstring medialis (HM), tibialis anterior, and gastrocnemius medialis (GM). The measurement was performed when the athletes arrived at the sports facility, before the warm-up and kicking. During the measurement, the athlete was lying down and stationary. The device probe was placed perpendicular to the midpoint of the muscle. The device was sufficiently pressed to the muscle, and short stimulations were applied to the tissue. The tone (Hz), elasticity (arb), and stiffness (N/m) values appearing on the screen were noted. Three measurements were performed for each muscle and averaged. Higher values indicate higher muscle tone and stiffness and lower elasticity (Chang et al., 2024; Deniz et al., 2024).

EMG and IMU acquisition

A wireless-mobile (Android application) electromyography (EMG) system (TRIGNO, Delsys Inc., Natick, MA, USA, input impedance <10 ohms, baseline noise <750 nV root mean square (RMS), effective EMG signal gain 909 V/V ± 5%) was used to obtain raw EMG data for the RA and RF. EMG signals were sampled at 2,000 Hz and a bandwidth of 20–450 Hz, full-wave rectified and smoothed with a second-order Butterworth low-pass filter with a cut-off frequency of 6 Hz. The distance between the metal surfaces of the wireless electrode was 1 cm. The Surface Electromyography for the Non-Invasive Assessment of Muscles recommendations were considered in the preparation of the skin and placement of the wireless electrodes (Hermens et al., 2000). For RA the electrode was placed 3 cm above and 2 cm lateral to the umbilicus, over the muscle belly and parallel to the muscle fibers (Clark, Holt & Sinyard, 2003). For RF the electrode was placed at the midpoint of the line from the anterior spina iliaca superior to the superior part of the patella, parallel to the muscle fibers (Fig. 2) (Hermens et al., 2000). The RF—the superficial muscle showing the highest activation along with the iliopsoas muscle (Brophy et al., 2007)—and the RA—one of the core muscles that may affect the thigh rotational velocity during instep kick, with a possible force transfer from trunk to leg, along with the anterior superficial fascia (Wilke et al., 2016)—were preferred for EMG analysis.

Figure 2 The placement points of the EMG and IMU electrodes.

EMG, Electromyography; IMU, inertial measurement unit. Before data recording, the EMG+IMU sensor was securely fixed to the thigh with adhesive tape.

The mean RMS collected from the muscles during the backswing and acceleration phases were normalized according to the maximum activation obtained during the instep kick (%maximum activation = muscle activity (RMS)/maximum activation of the muscle × 100). To reduce the possibility of obtaining normalized EMG levels greater than 100%, a normalization method with high reliability was used (intraclass correlation coefficient > 0.80) (Albertus-Kajee et al., 2011; Halaki & Ginn, 2012; Meghdadi, Yalfani & Minoonejad, 2019).

Two-dimensional video analysis was conducted to determine temporal characteristics of the kicking motion. A high-resolution video camera (Galaxy-S22, Samsung, Suwon, Korea) was positioned on a tripod at a height of approximately 1.5 m, located approximately 6 m directly perpendicular to the athlete’s sagittal plane (Deniz et al., 2024). It recorded the instep kick at a rate of 240 frames per second (fps). The camera and EMG data acquisition program (EMG Works® version 4.8.0; Delsys Inc., Natick, MA, USA) were synchronized using an LED system. To achieve precise synchronization between the EMG signals and the video-based kinematic data, a custom Android application was developed using App Inventor. This application controlled an LED light via an ESP32 microcontroller. At the beginning of each trial, the high-speed video camera was activated first. Subsequently, upon initiating EMG recording, the LED emitted a brief flash that was clearly visible in the video footage. This light flash served as a temporal marker, allowing accurate alignment of the EMG and kinematic data streams during post-processing. To support the temporal accuracy of these phase transitions, changes in movement direction were also tracked using gyroscopic velocity data from the inertial measurement unit (IMU; the sensor described in detail below) synchronized with the EMG system.

The recording of the kick was separated into the backswing and acceleration phases, in which the highest kinematic and kinetic values were expected (Brophy et al., 2007), using Kinovea (version 0.8.15, Kinovea Open Source Project) software that permitted single-frame viewing. Temporal calibration was performed in Kinovea software using the “Video → Configure video timing” function to align the playback frame rate with the actual capture frame rate; however, no spatial calibration was required since no distance measurements were extracted from the video. The backswing phase demonstrates the temporal window between the toe off of the kicking leg and the maximum hip extension, and the acceleration phase shows the temporal window between the maximum knee flexion and the ball strike (Brophy et al., 2007). The start and end times of each phase were marked on the video records. The EMG and IMU data for these periods were collected.

Trigno Avanti sensor-IMU Technology® (Delsys, Boston, MA, USA) was used to evaluate the thigh rotational velocity and acceleration during the instep kick. The sensor was placed on the midpoint of the thigh (Fig. 2) and fixed with adhesive tape. To minimize the influence of soft tissue oscillations on the IMU sensor, the sensor was securely affixed to the thigh using an adhesive band, which was wrapped circumferentially around the limb. Furthermore, during the analysis of IMU signals, the rotational velocity waveforms of the thigh were examined to confirm the presence of a linear peak pattern. The IMU sensor measured the rotational velocity (degree per second (deg/s)) and acceleration (g force) of the thigh along the kick in three planes (frontal, sagittal, and horizontal) (Deniz et al., 2024). The peak rotational velocity and acceleration values in three planes were considered for the data analysis.

Data analysis

Statistical analyses were carried out using SPSS (version 27.0; SPSS, Armonk, NY, USA). The distribution of data was determined by visual (histograms and probability plots) and analytical (Shapiro–Wilk test) methods. Pearson’s correlation test was used to determine the potential relationship between the biomechanical parameters and the maximum ball speed. Variables that were significant at the p < 0.05 level in the univariate analysis were included in the multiple linear regression analysis. Prior to regression analysis, the assumptions of multiple linear regression were assessed. Multicollinearity was evaluated using variance inflation factor (VIF < 5) and tolerance values (>0.1). Linearity, homoscedasticity, and normality of residuals were examined visually using scatterplot and histogram analyses. Alpha values of <0.05 were considered statistically significant. Graphs illustrating the results of the multivariate analysis were created using GraphPad Prism 10.0 software (GraphPad Software, LLC, San Diego, CA, USA).

A priori power analysis was not conducted to determine the sample size. However, a post hoc power analysis was conducted to calculate the power of the univariate analyses by using G*Power Software (version 3.0.18; Heinrich-Heine-Universität Düsseldorf, Germany).

Results

The mean value of the maximum ball speed was 17.3 ± 2.3 m/s. Tables 2 and 3 shown the results of the correlation analysis between the maximum ball speed and the biomechanical parameters. Significant negative associations existed between maximum ball speed and HM (Pearson’s r = −0.656; effect size = 0.43; power > 0.95) and GM elasticity (Pearson’s r = −0.680; effect size = 0.46; power > 0.95). The negative correlations indicate that increased HM and GM elasticity are associated with reduced maximum ball speed. The negative correlations indicate that lower HM and GM elasticity values—which reflect higher muscle elasticity—are associated with greater maximum ball speed (Table 2). Furthermore, significant positive correlations (Pearson’s r ranging from 0.380 to 0.619 and effect sizes ranging from 0.14 to 0.38) were found between the maximum ball speed and RA–RF activation, the maximum thigh rotational velocity in the sagittal plane, and the maximum thigh rotational acceleration in the sagittal and horizontal planes during the acceleration phase (p < 0.05 for all). These positive correlations suggest that greater RA–RF muscle activation and higher thigh rotational velocity and acceleration contribute to increased maximum ball speed (Table 3).

Table 2 Correlation of maximum ball speed with muscle tone, elasticity, and stiffness.

	Tone (Hz)	Elasticity (arb)	Stiffness (N/m)	
Muscle	Mean ± SD	r	p	Mean ± SD	r	p	Mean ± SD	r	p	
RA	14.23 ± 1.15	−0.133	0.477	0.75 ± 0.13	−0.179	0.334	223.02 ± 34.19	−0.131	0.481	
GMed	15.36 ± 0.68	0.064	0.733	1.00 ± 0.09	0.160	0.389	252.36 ± 23.57	0.020	0.915	
RF	13.09 ± 0.74	0.235	0.203	0.96 ± 0.11	−0.014	0.939	191.76 ± 29.27	0.231	0.212	
Amag	14.38 ± 0.64	−0.084	0.652	1.00 ± 0.12	0.038	0.840	233.47 ± 15.41	−0.139	0.457	
HM	18.11 ± 1.10	−0.048	0.797	0.84 ± 0.09	−0.656	<0.001	347.02 ± 34.07	0.022	0.904	
TA	14.71 ± 1.18	−0.024	0.899	1.04 ± 0.15	0.336	0.064	240.14 ± 20.65	0.131	0.483	
GM	13.03 ± 1.17	−0.122	0.512	1.13 ± 0.14	−0.680	<0.001	194.77 ± 37.19	−0.173	0.351	
Note:

Arb, Relative arbitrary unit; SD, standard deviation; RA, rectus abdominis; GMed, gluteus medius; RF, rectus femoris; Amag, adductor magnus; HM, hamstring medialis; TA, tibialis anterior; GM, gastrocnemius medialis. Significant associations (p < 0.05) are highlighted in bold.

Table 3 Correlation of maximum ball speed with muscle activity and thigh rotational velocity and acceleration.

	Backswing phase	Acceleration phase	
	Mean ± SD	r	p	Mean ± SD	r	p	
EMG (%)							
Rectus abdominis	54.20 ± 19.45	0.190	0.307	64.11 ± 22.01	0.494	0.005	
Rectus femoris	51.37 ± 20.68	−0.107	0.566	69.46 ± 18.31	0.579	<0.001	
VelocityMAX (deg/s)							
Frontal plane	842.79 ± 292.37	0.271	0.140	615.72 ± 240.76	0.299	0.103	
Sagittal plane	1,099.24 ± 339.94	0.294	0.109	975.35 ± 400.60	0.619	<0.001	
Horizontal plane	666.50 ± 314.46	0.223	0.227	410.72 ± 247.43	0.104	0.579	
AccelerationMAX (g)							
Frontal plane	10.46 ± 4.15	0.034	0.854	7.37 ± 3.03	0.237	0.198	
Sagittal plane	11.85 ± 4.44	0.352	0.052	8.00 ± 3.96	0.435	0.014	
Horizontal plane	12.11 ± 4.36	0.265	0.150	9.28 ± 4.45	0.380	0.035	
Note:

SD, Standard deviation; EMG, electromyography; Max, maximum, deg/s, degree per second; g, acceleration of gravity. Significant associations (p < 0.05) are highlighted in bold.

Biomechanical parameters were significantly associated with the maximum ball speed in the univariate correlation analysis and thus included in multiple linear regression analysis with two models. The multiple linear regression analysis demonstrated that important factors for maximum kick velocity were HM (B = −36.84) and GM (B = −26.83) elasticity and the maximum thigh rotational velocity in the sagittal plane during the acceleration phase (B = 0.01) (adjusted R2 = 0.56, Delta R2 = 0.17) (Table 4, Fig. 3).

Table 4 Multiple linear regression analysis for the association between biomechanical parameters and maximum ball speed.

Model 1	B (SE)	ß	p	
GM-Elasticity	−26.83 (8.13)	−0.465	0.003	
HM-Elasticity	−36.84 (12.46)	−0.417	0.006	
Model 2	
GM-Elasticity	−19.68 (7.24)	−0.342	0.012	
HM-Elasticity	−11.62 (13.40)	−0.132	0.395	
RA-EMGAcc	0.09 (0.05)	0.249	0.060	
RF-EMGAcc	0.05 (0.07)	0.103	0.524	
Velocity-SaggittalAcc	0.01 (0.00)	0.299	0.034	
Acceleration-SaggittalAcc	0.48 (0.33)	0.231	0.162	
Acceleration-HorizontalAcc	−0.12 (0.30)	−0.065	0.698	
Notes:

R2 = 0.56 for model 1.

ΔR2 = 0.17 for model 2.

Constant = 123.61 for model 1.

Constant = 76.39 for model 2.

GM, gastrocnemius medialis; HM, hamstring medialis, RA-EMGAcc, rectus abdominus activation in acceleration phase; RF-EMGAcc, rectus femoris activation in acceleration phase; Acc, acceleration phase; B, unstandardized coefficients; SE, standard error; β, standardized coefficients; R2, adjusted R-squared. Significant associations (p < 0.05) are highlighted in bold.

Figure 3 Multiple linear regression analysis for the hamstring medialis/gastrocnemius medialis and maximum ball speed.

GM, gastrocnemius medialis; HM, Hamstring medialis.

Discussion

This study investigated the biomechanical factors in prepubescent male soccer players associated with the development of high ball speed. It was revealed that HM and GM elasticity, along with the maximum thigh rotational velocity in the sagittal plane during the acceleration phase, were the most important determinant parameters of maximum ball speed. RA and RF activity as well as thigh rotational velocity and acceleration, were also associated with the maximum ball speed. To the best of our knowledge, this is the first study to demonstrate an association between muscle elasticity and maximum ball speed in prepubescent athletes.

On a biomechanical basis, the stretch–shorten cycle of agonist and antagonist muscles during the instep kick likely affects ball speed (Amiri-Khorasani et al., 2012). For the stretch–shorten cycle to be optimal, the contraction abilities of the muscles and their viscoelastic properties must be at appropriate levels (Taylor et al., 1990). Avrillon et al. (2020) showed that hamstring elasticity is a key factor distinguishing specialized athletes from the general population. However, past studies investigating the determinants of ball speed have not focused on the viscoelastic properties of muscles (Brophy et al., 2007; Cerrah et al., 2011; Dörge et al., 1999; Kellis & Katis, 2007; Nunome et al., 2006). In the current study, multiple linear regression analysis revealed that HM and GM elasticity were the significant determinants of ball speed in prepubescent male soccer players. These muscles, located in the posterior thigh and leg, are connected by the posterior superficial fascia and control anterior movements of the hip, knee, and ankle. Additionally, as a biarticular muscle spanning both the ankle and knee joints, GM, along with HM, plays a role in coordinating knee extention. It is proposed that during the late acceleration phase of the instep kick, the elasticity of the HM and GM muscles—due to their tendency to return to their original length—creates a passive opposing force, which counteracts the action of the muscles acting as agonists for the instep kick. Therefore, in athletes with high HM and GM elasticity, the torque generated by the hip flexors and knee extensor muscles is more effectively transmitted to the ball through the foot, resulting in higher ball speeds. It is noteworthy, however, that despite a significant interaction between muscle elasticity and maximum ball speed, no interaction was observed between tonus and stiffness values. This result indicates that the muscle’s tendency to return to its original length after elongation affects maximum ball speed more than baseline tonus or the ability to elongate. Further research examining the effects of muscle elasticity, tonus, and stiffness on athletic performance will provide deeper insights.

Many previous studies have aimed to determine the relationship between the activation patterns of lower extremity muscles and the instep kick velocity (Brophy et al., 2007; Cerrah et al., 2011, 2024; Dörge et al., 1999; Kellis & Katis, 2007; Augustus et al., 2024; Naito, Fukui & Maruyama, 2010). Considering the results of these studies, the significant positive relationship between RF muscle activation, thigh rotational velocity, and the ball speed for a successful kick is not surprising. A high level of hip flexor and RF activation is required to generate excessive rotational torque (thigh rotational velocity in sagittal plane) during the acceleration phase (Brophy et al., 2007; Cerrah et al., 2024). In addition to the high activation levels of the thigh muscles, the energy transfer from the support leg and trunk muscles to the distal segments of the kicking leg is also associated with higher ball velocities (Augustus et al., 2024; Naito, Fukui & Maruyama, 2010). Augustus et al. (2024) demonstrated that energy transferred from the support leg and torso early in the kicking phase (during arc formation and leg cocking) enhances pelvis and kicking leg energies during the downswing, with the magnitude of these transfers being linked to faster foot velocities. In the current study, the significant positive correlation between RA activity during the acceleration phase and higher ball velocities can be considered within this context. RA, which generates force to stabilize the core during the instep kick, may also contribute to thigh motion through force transmission along the anterior superficial fascia (Ajimsha et al., 2022; Lin et al., 2021). Stabilization of the trunk and the transmission of force from proximal muscles to the leg appear to be key factors for achieving a successful high ball speed.

The instep soccer kick is characterized by joint motions in multiple planes (Kellis & Katis, 2007). Previous studies have shown that the rotational velocity of the combined movement pattern, formed by flexion and rotation of the thigh during the acceleration phase, has an important effect on ball speed (Dörge et al., 1999; Nunome et al., 2006). Similarly, the current study highlights the importance of high sagittal and horizontal rotational accelerations during the early phase of the acceleration phase to achieve high ball speeds. Additionally, the rotation of the thigh plays a role in orienting the kicking foot, enabling proper alignment of the anterior-medial dorsal aspect for optimal contact mechanics (Campo et al., 2009). It should be noted that (Nunome et al., 2006) suggested that initial rapid acceleration is followed by a controlled deceleration of the thigh towards the end of the phase, which is essential for optimizing the transfer of energy to the lower leg and ensuring effective foot-to-ball contact (Nunome et al., 2006).

The instep soccer kick is an open kinetic chain motion. The rotational velocity of the shank and, ultimately, the foot, along with the rotational velocity of the thigh, are critical determinants of maximum ball velocity (Kellis & Katis, 2007). In this study, proximal segments, particularly the thigh and hip, were primarily focused on due to their critical role in generating rotational torque during the instep kick. However, incorporating analyses of distal segments and their associated muscle groups could provide a more comprehensive understanding of the movement. Future studies should investigate the association between the activation and viscoelastic properties of muscles and shank-foot rotational velocity across broader age groups.

Limitations

This study has some limitations. First, muscle viscoelasticity was assessed in a stationary, supine or prone position, consistent with established methods in the literature (Chang et al., 2024; Pruyn, Watsford & Murphy, 2016). While this measurement provides valuable insights into the intrinsic properties of muscles, it remains unclear to what extent viscoelasticity assessments performed under static conditions apply to dynamic situations. Second, although the test setup was designed to approximate the natural instep kick pattern of the majority of participants, the maximum ball speeds and biomechanical parameters obtained in this study may not fully reflect those observed during match play or typical training situations. Third, EMG analysis of the deep abdominal (core) muscles could not be performed. Therefore, the potential force transmission from the deep abdominal muscles to the anterior thigh muscles during the instep kick could not be determined. Fourth, since the study included only athletes aged 11 to 12 years and the test was limited to three kicks, the findings may not be generalizable to all prepubescent male athletes or reflect full performance variability. Lastly, 3D analysis, considered the “gold standard” for kinematic evaluation (Dingenen et al., 2018), was not utilized for assessing thigh kinematics due to its high cost, the complexity of setup in field conditions, and the need for specialized expertise.

Conclusions

GM and HM elasticity and thigh rotational velocity in sagittal plane were the considerable biomechanical parameters affecting the maximum ball speed during instep kick among prepubescent male soccer players. Additionally, RA and RF activity and the rotational acceleration of the thigh in the sagittal and horizontal planes are associated with the maximal ball speed. Considering these results, it is recommended that comprehensive training programs for young athletes focus on improving both muscle elasticity and activation to optimize kicking performance.

Supplemental Information

Supplemental Information 1 Raw data.

We extend our gratitude to all participants who voluntarily contributed to this study.

Artificial intelligence (ChatGPT, OpenAI) was used solely to assist in the translation of a small portion of the manuscript. It was not used for the generation of scientific content or figures.

Additional Information and Declarations

Competing Interests

The authors declare that they have no competing interests.

Author Contributions

Volkan Deniz conceived and designed the experiments, performed the experiments, analyzed the data, prepared figures and/or tables, authored or reviewed drafts of the article, and approved the final draft.

Abdullah Kılcı conceived and designed the experiments, performed the experiments, authored or reviewed drafts of the article, and approved the final draft.

Human Ethics

The following information was supplied relating to ethical approvals (i.e., approving body and any reference numbers):

The Clinical Research Ethics Committee of Çukurova University approved the study (Decision no.: 143/75, Approval date: 5 April 2024).

Data Availability

The following information was supplied regarding data availability:

Raw data is available in the Supplemental Files.

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
