# Peer review of "Biomechanical determinants of high ball speed during instep soccer kick by prepubescent male athletes: the importance of muscle elasticity"

_PeerJ, doi:10.7717/peerj.19923_

## Round 0.1 · original submission · Major Revisions

The article must be carefully revised, considering all the reviewers' comments.

·

Basic reporting

First, about language: it is Unambiguous, and professional English is used throughout. At the same time, Literature references are correct, they have a sufficient field background/context provided, and literature is appropriately referenced. The structure, figures, and tables conform to a clear and acceptable format. The presented results are relevant to the content of the article, appropriately described and labeled. I am really satisfied with both the presentation and the display of results.

Experimental design

Looking at the experimental plan, my opinion is that the authors respected the methodological instructions and prepared and conducted the research well. They have well-explained scientific questions and problems, and have set the goals and hypotheses of their research well. They measured with relevant equipment and followed both the ethnic and literature-suggested methods of measurement and data analysis. They presented the results in a satisfactory manner and explained them results well.

Validity of the findings

The results obtained are expected. But this does not diminish the scientific contribution of this work. The presented results provide sufficient new data and explain kicking performance a little better. They explained the results well with their comments, but they also linked the comments to previous results from other studies. They also determined the limitations of the study and provided valuable recommendations for further research.

Reviewer 2 ·

Basic reporting

English is clear. References are in accordance with the research context. Background is adequate. The article is well-organized and clear. But the table data section needs to be improved according to the comments. In addition, the results are relevant to the hypothesis presented.

Experimental design

There is no explanation of the assumptions of linearity, heteroscedasticity, and multicollinearity. A good correlation should have a linear relationship between the independent and dependent variables, no multicollinearity in the regression model, and be free from symptoms of heteroscedasticity.

Validity of the findings

Some inconsistencies were noted between the reported data and the SPSS output, particularly in the mean ± SD, Pearson correlation results, constant values, regression coefficients (B), and p-values, which should be carefully rechecked.

Additional comments

As mentioned above, this text requires revision in the data analysis section, namely an explanation of the assumptions that must be met when using multiple linear regression and a revision of the data presented in the table.

Annotated reviews are not available for download in order to protect the identity of reviewers who chose to remain anonymous.

Reviewer 3 ·

Basic reporting

A clear and unambiguous English level has been used throughout the manuscript. However, the literature references do not cover sufficient studies, i.e., there is no mention of previous studies on the biomechanics of kicking in youth soccer players (e.g., https://pubmed.ncbi.nlm.nih.gov/21904274/; https://doi.org/10.1080/02640414.2010.504781). Figures and tables are, in general, of good quality. Hypotheses are too general (Lines 100 – 103). I recommend being as specific as possible regarding the direction of the correlations. In addition, insert the references (if applicable) used to formulate the hypotheses.

Experimental design

Despite fitting the aims and scope of the journal, the study is not too original, given that various previous studies have related kinematics and EMG to kicking performance. The authors should stress the original aspect of the work, for example, the investigation about the role of muscle viscoelastic properties in ball kicking, and improve the rationale. Based on that, the research question is partly relevant & meaningful. The research experiment seems rigorous, and the methods are generally well described; however, one major problem of the study is that a 2D kinematic analysis was adopted. In the case of soccer kicking, the use of such an approach is recognised to produce distorted data as compared to 3D (Nunone and Ikegami, 2006 - https://ojs.ub.uni-konstanz.de/cpa/article/view/135/96 ). The synchronization of the EMG with the kinematics data should also be better described in detail by the authors (Lines 171 to 173).

Validity of the findings

Regarding the data sharing, the data sheets used for statistical analysis were provided by the authors. However, data matrices such as those from EMG and kinematics time series raw data are not provided. Conclusions should also be revised because they are not limited to supporting results. This includes the statements reported as from Lines 332 to 336.

Additional comments

Based on the previous comments, my recommendation is that the work requires MAJOR REVISION, mainly attempting to improve the rationale (highlight the potentially innovative aspects of the study in comparison with previous literature), some aspects of methods, and avoid speculative conclusions.

Reviewer 4 ·

Basic reporting

The introduction is significantly long and could be condensed, focusing more on the specific research. If you are solely focusing on soccer players, there is no need to mention Australian Rules Football or throwing mechanics.

Please check your spelling carefully, as some mistakes were found.

Do you have an image of the EMG placement on the muscle bellies?

Experimental design

Can you please provide more details in the methodology section of this research such as the level of playing experience, surface type and conditions (e.g. natural grass or artificial grass – and provide explicit detail) as well as some indication as to where players were in their growth phase as an accelerated growth rate could influence biomechanics and thus kicking actions.
Was each kick timed? Or were players instructed to kick at a specific point? EMG activation during the acceleration phase would be significantly different if players were performing their kicks in their own time versus being told when to kick by a researcher.
Your constraints placed on the run-up, while an effort to minimize variability, seem heavily constricting, as most players, when performing an instep kick, will approach the ball with some angle. Also, in instances where the target was missed, what did you do? Was the test repeated?
Does the change in measurement position for myotonometry (e.g., lying down) accurately reflect what the muscles may be like in kicking?
Why not use the MVC method to report EMG activity? When using the kicking action to normalise the data, was the same kick used for each muscle? Or were different maximum values obtained from the three different kicks?

Validity of the findings

Your limitations also include the number, experience, and age of participants. Your research is titled “prepubescent male athletes” – however, the prepubertal age for males is generally considered 9-14. The number of kicks also seems to be a limitation. Giving players 3 kicks, with a run up and in conditions they are not accustomed to, especially considering the restrictions of the run up, seems unreasonable. Were any lower limb biomechanics recorded during a player's normal run-up? If so, were there significant differences?

Additional comments

I want to commend the authors on their work and the novel nature of the research. I do have some concerns about the methodology, particularly the participant details and the constraint of the run-up, as detailed below.

Overall comments:

The introduction is significantly long and could be condensed, focusing more on the specific research. If you are solely focusing on soccer players, there is no need to mention Australian Rules Football or throwing mechanics.

Can you please provide more details in the methodology section of this research such as the level of playing experience, surface type and conditions (e.g. natural grass or artificial grass – and provide explicit detail) as well as some indication as to where players were in their growth phase as an accelerated growth rate could influence biomechanics and thus kicking actions.

Was each kick timed? Or were players instructed to kick at a specific point? EMG activation during the acceleration phase would be significantly different if players were performing their kicks in their own time versus being told when to kick by a researcher.

Your constraints placed on the run-up, while an effort to minimize variability, seem heavily constricting, as most players, when performing an instep kick, will approach the ball with some angle. Also, in instances where the target was missed, what did you do? Was the test repeated?

Does the change in measurement position for myotonometry (e.g., lying down) accurately reflect what the muscles may be like in kicking?

Why not use the MVC method to report EMG activity? When using the kicking action to normalise the data, was the same kick used for each muscle? Or were different maximum values obtained from the three different kicks?

Your limitations also include the number, experience, and age of participants. Your research is titled “prepubescent male athletes” – however, the prepubertal age for males is generally considered 9-14. The number of kicks also seems to be a limitation. Giving players 3 kicks, with a run up and in conditions they are not accustomed to, especially considering the restrictions of the run up, seems unreasonable. Were any lower limb biomechanics recorded during a player's normal run-up? If so, were there significant differences?

Do you have an image of the EMG placement on the muscle bellies?

Individual Comments to authors:
Line 23 – Please define what you mean by “rotational velocity-acceleration.” Do you mean rotational velocity and rotational acceleration?
Line 24 and throughout – please clarify what you mean by ball speed – is it ball exit velocity from the kick? Or is it measured when the ball is freely travelling in the air?
Line 49-50 – Can you define ‘ball releasing velocity’? Do you mean ball exit velocity?
Line 59-61 – It would be good to describe what an instep kick is earlier in the introduction.
Line 74 – What is an academy soccer player? I assume you mean someone under the age of 18?
Line 119-120 – Please provide more details as to how these were recorded. Specifically, the location in proximity to the test site, and what the temperature was during testing.
Line 129 – What were the submaximal kicks? Instep or sidestep? Line 131 – Check spelling: ‘Each player kicks’
Line 137 – When was the ball speed measured specifically? Was it a certain point after impact?
Line 138- Line 141 – Please clarify if the radar gun needs to be 14m away to measure ball speed, or if the maximum distance it can measure from is 14m? And if so, is there any change in recording at different distances?
Line 174 – As there was only 1m distance for run up, was there a significant time difference between these two phases? The acceleration phase was surely quite short.
Line 254 – What age were these athletes? Does the claim still apply for 11 and 12-year-olds?
Line 266-268 – How did the changes in elasticity then correlate to the rotational acceleration of the leg? Did your results support this claim?
Line 272 – Check spelling ‘researchs’

---

## Round 0.2 · Minor Revisions

I think the authors can respond quickly to what was asked. Some comments are already in the article, and the authors just need to organize the concepts in the article's structure. The maximum values ​​of the analyzed variables occurred in the acceleration phase, from what I could observe. Therefore, it was not clear whether it was necessary to separate the kick into two phases, and perhaps the kinematic analysis is not necessary. I ask the authors to comment this issue.

Reviewer 2 ·

Basic reporting

The data presented in the tables has been corrected according to comments, the results match the SPSS output, and the use of English is clear.

Experimental design

The assumptions of linearity, heteroscedasticity, and multicollinearity as prerequisites for the regression model have been well explained in the data analysis section

Validity of the findings

Improvements in the research results data section are in accordance with SPSS output such as mean ± SD, Pearson correlation results, constant values, B values, and p values. Thus increasing the accuracy of statistical findings.

Additional comments

As mentioned above, this text has been revised in the data analysis section, especially the explanation of the assumptions of the multiple linear regression model and the presentation of table data is in accordance with the SPSS output. In conclusion, this manuscript is accepted.

Reviewer 3 ·

Basic reporting

I would like to thank the authors for taking into account my comments in the revised manuscript. I have still only a major concern regarding the procedure to evaluate kicking kinematics. In this sense, I strongly suggest to the authors to insert more information about the validity of the kinematic measurements, including as for example (i) measurement error for the calibration and reconstruction of 2D data; (ii) the type of tracking (i.e. automatic, semi-automatic, manual?) and associated intra- and inter-rater consistency, if applicable (iii) which dependent variables were extracted. If these measures are reported by the authors, then the article may have acceptable standards, otherwise the method becomes unclear and do not allow future replication of the study. Thank you.

Experimental design

no comment

Validity of the findings

no comment

Reviewer 4 ·

Basic reporting

The condensing of the introduction is good and focusses directly on the research at hand, as well as specific gaps in the research. I think it could be further condensed, particularly lines 82-104 as I feel it repeats a lot of what was said in the previously paragraph, especially lines 101-104. In lines 92 and 93 can you please define what you mean by backward and forwards phases of the instep kick? Later on you have described them as backswing and acceleration which seems a more appropriate label but perhaps a diagram would be good to describe the different phases?

Experimental design

Did you include a variety of position players within your recruitment process? I am still struggling with your restrictions on run up for this test. In line 147-150, you say there is “considerable variation” in the approach angle. Yet in the next line you state a restriction of 30-45 degrees “as the approach which most participants self-selected”. 2-3 steps is also a small run up distance if instructed to perform a maximal speed kick.
Can you find a more appropriate reference of validating myotonometry in young athletes? The reference provided (Pruyn et al, 2016) recruited athletes aged 18-35, which is significantly different from the 11-12 year olds used in your study.

Validity of the findings

What were the variations in kinematics between each kick for each participant? Were these significant? As I mentioned previously, playing athletes in a new kicking environment and having them make 3 successful kicks, at maximum speed, would surely lend itself to a great deal of variation in biomechanics. Especially when the previous 5 warm up kicks were not controlled.

Additional comments

Overall, the updated manuscript shows significant improvement, however I would like more detail regarding the specific variations within each players kick.

---

## Round 0.3 · Minor Revisions

Please, I ask the authors to quickly revise what Reviewer #4 has requested. Thanks.

Reviewer 3 ·

Basic reporting

Thank you for the revisions

Experimental design

no comment

Validity of the findings

no comment

Reviewer 4 ·

Basic reporting

The reporting and format of the paper is of good quality, however please avoid the use of personal language such as "we". Such examples can be found on lines 132, 135, 324, 333, 371 and 400.

Experimental design

Appropriate changes have been made and limitations highlighted

Validity of the findings

Appropriate changes have been made to this section.

Additional comments

Please address minor grammatical changes as requested.

---

## Round 0.4 · accepted · Accept

The authors have carefully revised the manuscript in accordance with all Reviewers' comments. The manuscript is ready for publication.